# Assessing the Healthy Food Partnership’s Proposed Nutrient Reformulation Targets for Foods and Beverages in Australia

**DOI:** 10.3390/nu12051346

**Published:** 2020-05-08

**Authors:** Emalie Rosewarne, Liping Huang, Clare Farrand, Daisy Coyle, Simone Pettigrew, Alexandra Jones, Michael Moore, Jacqui Webster

**Affiliations:** The George Institute for Global Health, The University of New South Wales, Sydney, NSW 2006, Australia; hliping@georgeinstitute.org.au (L.H.); cfarrand@georgeinstitute.org.au (C.F.); dcoyle@georgeinstitute.org.au (D.C.); spettigrew@georgeinstitute.org.au (S.P.); ajones@georgeinstitute.org.au (A.J.); mimomph@gmail.com (M.M.); jwebster@georgeinstitute.org.au (J.W.)

**Keywords:** reformulation, food supply, Australia

## Abstract

Non-communicable diseases (NCDs) are the leading cause of mortality and morbidity worldwide. Unhealthy diets are one of four main behavioral risk factors contributing to the majority of NCDs. To promote healthy eating and reduce dietary risks, the Australian Commonwealth Government established the Healthy Food Partnership (HFP). In 2018, the HFP consulted on proposed nutrient reformulation targets for 36 food categories to improve the overall quality of the food supply. This study assessed whether the proposed targets were feasible and appropriate. The HFP used a five-step approach to inform the proposed targets. We replicated and extended this approach using a different nutrient composition database (FoodSwitch). Products in FoodSwitch were mapped to the proposed HFP targets. The proportion of products meeting each target was calculated and the FoodSwitch data were compared with HFP data to determine whether the proposed target nutrient levels were appropriate or whether a more stringent target was feasible. Products from the FoodSwitch database (10,599) were mapped against the proposed HFP categories: 8434 products across 30 categories for sodium, 2875 products across seven categories for sugar, and 612 products across five categories for saturated fat. The analyses revealed that 14 of 30 proposed HFP targets for sodium, one of seven targets for sugar, and one of five targets for saturated fat were feasible and appropriate. For the remaining 26 reformulation targets, the results indicate that these target levels could be more stringent and alternative targets are proposed. The draft HFP targets are feasible but the majority are too conservative. If Australia is to meet its commitment to a 30 per cent reduction in the average population salt intake by 2025, these targets could be implemented as interim targets to be reached within two years. However, the opportunity exists to improve the food supply and strengthen the HFP’s population health impact by adopting more ambitious and incremental targets. Reformulation programs should be prioritized and closely monitored as part of a coordinated, multi-faceted national food and nutrition strategy.

## 1. Introduction

Non-communicable diseases (NCDs) are the leading cause of mortality and morbidity worldwide [1]. NCDs are largely preventable, with the majority attributable to four behavioral risk factors, one of which is unhealthy diets [2]. Unhealthy diets increase the risk of many NCDs including cardiovascular diseases (CVDs), diabetes, and some cancers [3,4], through a myriad of metabolic changes, such as raised blood pressure, blood glucose and blood cholesterol levels, and overweight and obesity [2,4].

As part of a comprehensive policy response to promoting healthier diets, the World Health Organization (WHO) recommends countries adopt a range of policy actions to improve the food environment and make progress towards voluntary global NCD targets, such as a 30% reduction in salt intake by 2025 [5,6]. These include implementing programs to encourage reformulation of food products to reduce levels of nutrients associated with NCD risk and can lead to reduced population intakes of these adverse nutrients [5,7]. Salt reduction in particular is recognized as a ‘best buy’ strategy, being one of the most cost-effective and feasible interventions to implement [8], and there are technical and action packages to assist countries with implementation [9]. These reformulation initiatives and other food environment interventions can have population-wide impacts independent of individual behavior change [10,11].

As elsewhere, dietary risks are a leading mortality risk factor in Australia [1]. Population salt intake is almost double [12] the recommended maximum amount of 5 g/day [13], and population sugar and saturated fat intakes also exceed the recommended maximum amount of 10% total energy intake per day [14,15] (10.9% and 11.5% respectively [16,17]). In view of this, governments around the world, including the Australian Commonwealth Government, have established nutrient reformulation targets to reduce levels of sodium, sugar and saturated fat in the food supply. Previous voluntary targets in Australia resulted in sodium reductions in three product categories, bread, breakfast cereals and processed meat [18,19], however limited progress towards these targets was demonstrated overall [20]. In 2015, the Commonwealth Government announced a new initiative, the Healthy Food Partnership (HFP) [21]. Three years later, in 2018, the HFP’s Reformulation Working Group proposed a set of nutrient reformulation targets including 30 sodium, seven sugar and five saturated fat targets for 36 subcategories of foods and beverages, which were informed by nutrient data from the FoodTrack database [22].

The Reformulation Working Group released the proposed targets for public consultation in July 2018, supported by a detailed rationale document [23]. The objective of the consultation was to obtain stakeholder views on the feasibility of the proposed targets, and the appropriateness of the food category definitions and implementation period [24]. The objective of the present study was to replicate and extend the target setting process undertaken by the HFP but using the FoodSwitch food composition database, which contained around 63,000 products from 2015 to 2017 and covered an estimated 90% of the market share, and compare the findings with the FoodTrack data published in the HFP rationale document [23]. This was to determine whether the proposed targets were feasible and appropriate or if there is scope for the introduction of more stringent targets and expansion of the targets to additional food categories. Improvements in the nutrient content of commonly consumed processed foods through food reformulation can reduce the burden of diet-related diseases, so it is important that the potential of this public health intervention is fully realized.

## 2. Materials and Methods

In this study, we assessed the HFP’s proposed nutrient reformulation targets by replicating and extending on their approach using a different food composition database (FoodSwitch). Here we describe the data source and exclusion criteria, as well as the process of mapping products in the FoodSwitch database to the HFP targets. We also discuss the process for assessing if each draft HFP target was feasible and appropriate, and determining the new target where there was scope for further reductions.

### 2.1. Approach Used by the HFP to Develop the Proposed Targets

The HFP working group considered five factors when setting the proposed target for each food category [23]. These were: (1) the type of target (absolute levels or percentage reduction); (2) existing and relevant targets nationally and internationally; (3) Health Star Rating (HSR) baseline nutrient cut-points for sodium, sugar and saturated fat from Australia and New Zealand’s voluntary front-of-pack nutrition labelling system [25]; (4) technical and safety limitations; and (5) existing means and ranges of nutrient levels of each food category from the FoodTrack data (2015−2017). Targets were generally proposed at a level where about one-third of the products in the FoodTrack database were already at or below the target.

### 2.2. Data Source

The Australian FoodSwitch Database contains the nutrient information of packaged foods and beverages from three sources: annual data collections in four large Australian supermarkets, crowd-sourced data from users of the FoodSwitch app and data provided by the food industry [26,27]. Data for foods and beverages are collected from the nutrition information panel and packaging through a series of photographs. More detailed information about this process is described in Dunford et al. [28]. We extracted FoodSwitch data for the period 1 January 2015 to 31 December 2017 to ensure our methods were comparable to the HFP analyses.

### 2.3. Exclusion Criteria

Duplicate products in the FoodSwitch database were not included. Duplicate products have the same barcode and product identification and only appear once in the database extract. We checked for any additional duplicate products (such as the same product in multiple pack sizes) and these were excluded based on three levels of matching: (1) product name and key nutrients (energy, carbohydrate, total fat, saturated fat, sodium, sugar, protein, and fiber); (2) brand identification and ingredients; (3) brand identification category and key nutrients (energy, carbohydrate, total fat, saturated fat, sodium, sugar, protein, and fiber).

### 2.4. Product Mapping to Reformulation Targets

Products in the FoodSwitch database were included in the analysis if they could be mapped to a proposed sodium, sugar or saturated fat target using the HFP food category definition and inclusion/exclusion criteria outlined in the rationale document [23]. For a supplementary analysis designed to assess the potential for expanding the HFP targets to cover additional food and beverage categories, products in the FoodSwitch database were mapped to the UK salt and sugar targets [29,30]. The UK salt targets are some of the earliest and most comprehensive incremental salt reduction targets, initially covering 85 food categories, and more recently, the UK have set sugar targets for 14 categories [29,30].

### 2.5. Data Analysis

Figure 1 outlines the process for assessing if each draft HFP target was feasible and appropriate, and determining the new target where there was scope for further reductions. We did this by performing five main analyses for each food category. We compared:The proportion of products in the FoodSwitch database already at or below the proposed HFP targets for sodium, sugar and saturated fat, to the HFP criteria for feasibility and appropriateness (i.e., approximately one-third of products (33rd percentile) already meeting the target [23]). A difference of less than 10% was used as a cut-point to indicate the data were similar, and a difference of greater than 10% (<23.3% or >43.3%) was considered different.The FoodSwitch 33rd percentile nutrient levels for sodium, sugar and saturated fat levels to the proposed HFP target nutrient levels. A difference of less than 10% was used as a cut-point to indicate the data were similar, and a difference of greater than 10% was considered different.The mean and range for each food category using FoodSwitch data to the mean and range using FoodTrack data. A difference in means of less than 10% was used as a cut-point to indicate means were similar, and a difference of greater than 10% was considered different.The HFP proposed targets to the UK salt and sugar targets [29,30].The HFP proposed targets to the Health Star Rating (HSR) baseline cut points for sodium, sugar and saturated fat [25].

The primary trigger for a recommended lower target was the proportion of products in the FoodSwitch database already meeting the proposed targets, while the other four criteria were used as checks that either supported or contraindicated the apparent need for new targets. A new target was proposed if more than 43.3% of FoodSwitch products were already at or below the target nutrient level (>10% difference to the HFP criteria for feasibility and appropriateness of approximately one-third of products (33rd percentile) already meeting the target). We considered revising the target if between 23.3% and 43.3% of FoodSwitch products were already at or below the target (<10% difference to HFP criteria). The target was deemed feasible and appropriate if it was within this range and set at a HSR cut-point, however a new target was recommended at the closest HSR cut-point (where feasible and appropriate) if the target was not already set at a HSR cut-point. New targets were proposed by considering: FoodSwitch 33rd percentile, mean and range; the UK targets; and HSR cut-points.

## 3. Results

There were 62,802 unique barcodes in the FoodSwitch database across the three years. After removing duplicates, 41,305 products remained. In total, 10,605 products were able to be mapped to at least one HFP target, and six products were excluded for missing sugar values. The final dataset comprised 10,599 products: 8434 products were mapped to the 30 proposed HFP food categories for sodium, 2875 products across the seven food categories for sugar, and 612 products across the five food categories for saturated fat. In the following sections, we provide results for each of the five analysis steps and describe new proposed targets based on FoodSwitch data.

### 3.1. Proportion of Products Already Meeting the Proposed HFP Targets

In 23 of 30 categories with sodium targets, six of seven categories with sugar targets, and four of five categories with saturated fat targets, the HFP criteria for feasibility and appropriateness of one-third of products already at or below the proposed HFP target was met. In fact, for 12 of 30 categories with sodium targets, four of seven categories with sugar targets, and four of five categories with saturated fat targets, at least 43.3% of products were already at or below the proposed HFP targets, a greater than 10% difference to the HFP criteria. The proportion of products in each category already meeting the proposed HFP targets varied greatly from 24% to 89% for sodium, from 41% to 67% for sugar, and from 31% to 72% for saturated fat (Table 1, Table 2 and Table 3).

### 3.2. 33rd Percentile Nutrient Level of the FoodSwitch Data Compared to Proposed HFP Targets

The FoodSwitch 33rd percentile nutrient level was similar to the proposed HFP target for 17 of 30 sodium categories, two of seven sugar categories and one of five saturated fat categories. The percent differences ranged from −96% to 11% for sodium, −36% to 0% for sugar, and −31% to 1% for saturated fat (Table 1, Table 2 and Table 3).

### 3.3. Comparison of Means between FoodSwitch and FoodTrack Data

FoodSwitch and FoodTrack means were similar for 12 of 30 sodium categories, five of seven sugar categories, and two of five saturated fat categories. Percent differences in means between the FoodSwitch and FoodTrack data ranged from −55% to 96% for sodium, −40% to 3% for sugar, and −12% to 29% for saturated fat. The variability in means and ranges is displayed in Table 1, Table 2 and Table 3.

### 3.4. Comparison between Proposed HFP Targets and UK Targets

The proposed HFP targets were set lower than the UK targets for 18 out of 30 sodium categories, equal to for three categories, and higher than the remaining nine categories. Due to the nature of the proposed sugar targets, including a mixture of percentage reductions and absolute sugar level targets, it was difficult to compare the UK and proposed HFP targets. However, it appeared that the HFP targets were well above the UK targets for breakfast cereals, above for yoghurts, and similar for muesli bars (Table 1, Table 2 and Table 3).

Based on the full list of food categories included in the UK targets, Australian targets could be set for at least 35 more food categories for sodium and at least 10 further food categories for sugar. Differences in category definitions between the HFP and UK targets resulted in 10 UK categories being partially covered by sodium targets and two UK categories being partially covered by sugar targets (Appendix A).

### 3.5. Comparison of Proposed HFP Targets to HSR Baseline Cut-Points

Of the proposed HFP targets, 23 of 30 sodium targets, three of seven sugar targets, and four of five saturated fat targets were set at HSR cut-points. In our analysis, we determined that it was feasible for 28 of 30 sodium targets, five of seven sugar targets, and five of five saturated fat targets to be set at HSR cut-points (Table 1, Table 2 and Table 3).

### 3.6. Proposing New Targets Based on FoodSwitch Data

Based on the above analysis, we propose new targets for 16 of 30 categories with HFP sodium targets, six of seven categories with HFP sugar targets, and four of five categories with HFP saturated fat targets (Table 1, Table 2 and Table 3). New targets were proposed for 12 of 30 categories with sodium targets, four of seven categories with sugar targets, and four of five categories with saturated fat targets as 43.3% of products were already meeting the proposed HFP target. The sodium target for savory sauces was the only exception. This target was not revised as the definition of the product category was deemed too broad to make an appropriate recommendation. In addition, four new sodium targets and two new sugar targets were proposed, as between 33.3% and 43.3% of products already met the target and the proposed target was not set at a HSR cut-point. These targets were set at the next lowest HSR cut-point, except ready meals (as the next lowest cut-point was likely infeasible) and dairy alternatives (where no lower cut-point exists).

## 4. Discussion

The present study is based on almost 10,600 packaged foods from the FoodSwitch database mapped against the proposed HFP targets to assess the extent to which these targets would drive food and beverage reformulation. Our analysis revealed that 26 of 42 proposed reformulation targets were too conservative. For most sodium targets, and almost all sugar and saturated fat targets, at least one-third (33.3%) of products, and in many cases more than 43.3% of products were already meeting the proposed HFP targets. This suggests that the potential impact of the HFP initiative would be limited by lenient target setting. The proposed HFP targets could be viewed as short-term interim targets to be reached within two years. However, the new, more ambitious targets we have proposed using FoodSwitch data would likely have greater impact on the food supply and population diets if implemented successfully, and accelerate Australia’s efforts towards achieving the global NCD targets, such as a 30% reduction in population salt intake [5]. New plans should also be made to include further step-wise reduced targets, as in the UK [31], for continued food supply improvements.

The dataset used by the HFP may have limited the appropriateness of the proposed target setting. For the three years of data included in the study, the FoodTrack database contained only 16,000 products from four major supermarkets in Australia [22]. By comparison, the FoodSwitch database contained almost 63,000 products for the same period and covers approximately 90% of the market share. The considerable difference in database size likely contributed to the greater than 10% difference in means for the majority of food categories. In addition, the FoodSwitch data indicated almost half of the targets were set at a nutrient level where more than 43.3% of products were already meeting the targets, 10% higher than the HFP suggested feasibility criteria. Further to this, for some categories it appeared that the FoodTrack data only contained one product (pesto and frankfurts/saveloys categories), while the much larger FoodSwitch database comprised at least 30 products in every category. By replicating the approach undertaken by the HFP using a larger database with greater market coverage, it was revealed that the majority of food category targets could feasibly be set lower than the HFP proposed targets. This illustrates the importance of utilizing a comprehensive food composition database for setting feasible and appropriate nutrient reformulation targets that will have greater impact on the food supply and population health.

This research also highlights the scope for extending the HFP reformulation targets beyond the existing proposed food categories. The HFP constrained target setting to categories that contributed more than one percent to the intake of that nutrient for the general population, or additionally considered if they contributed greater than one percent to (1) children’s diets or (2) high sodium consumers. Targets were not set for food categories high in a particular nutrient yet contributing less to dietary intake, nor for food categories that contribute substantial amounts of sodium, sugar or saturated fat to population subgroups diets. For example, instant noodles are high in sodium [32], yet contribute less than one percent to daily sodium intake in the general Australian population [33]. An American study showed instant noodle consumption varied by ethnicity, income, age and gender, with some population subgroups consuming a significantly higher quantity than others [34], and this is likely also the case in Australia. For reasons such as this, we propose that it would be appropriate to set nutrient targets for many additional food categories. In our analyses, we compared the proposed HFP targets to the current UK salt and sugar targets and provided evidence that there is scope to increase the number of food categories with a target in Australia. The HFP’s approach resulted in 30 sodium targets and seven sugar targets being proposed, compared to the much higher 76 and 14 categories included in the UK salt [30] and sugar [29] strategies, respectively. While the HFP undertook a logical approach to selecting categories, it also limited the scope and potential public health impact of the targets, and there is opportunity to set further targets and increase the number of products covered.

Modelling of the impact of the HFP targets reveals that further work will be needed to improve the food supply and population diets, and the new targets proposed in this study are an initial step. The HFP estimated that full compliance with the proposed targets would lead to nutrient intake reductions of 212 mg sodium (8.7%), 1.3 g total sugar (1.2%) and 0.24 g saturated fat (0.9%) per person per day [23]. These estimates were based on dietary intake data from the 2011−2012 National Nutrition and Physical Activity Survey (NNPAS), which has known limitations, such as the 24-h recall data being self-reported and some dietary sources being missed [33]. For example, the NNPAS data underestimated population sodium intake by almost 40% compared to the best estimate from 24-h urine samples [12,33], which may have impacted modelling estimates. Further research is needed to model nutrient intake reductions and health outcomes based on the new proposed targets, and potentially additional scenarios, using FoodSwitch and sales-weighted data, and using more reliable and recent dietary intake measurements. To further increase the potential impact of the initiative, targets should also be established for the out-of-home sector, including fast food, takeaway, restaurant and café foods [23]. While voluntary pledges for energy, sodium and fats/oils reduction for this sector have been developed [35], they lack detail as to the level of reformulation to be achieved. With out-of-home sector purchases continuing to grow [36], establishing maximum nutrient targets for this sector, in addition to commercial packaged food products, is vital to achieve the reductions in nutrient intakes needed to meet global targets, such as the 30% salt reduction target by 2025 [5].

Nutrient reformulation targets are an important first step in improving the diet of Australians. However, robust monitoring and evaluation frameworks and systems are needed to hold industry and government accountable, track progress against the targets and help identify challenges early [26,37]. In February 2020, the HFP endorsed the first wave of targets and announced intentions to review the remaining targets in the next few months. Whilst an implementation plan is yet to be agreed, the consultation documents suggested that the targets should be reached within four years, with self-reported industry progress reports at two and four years [23]. In view of the government’s commitment to achieving the global NCD targets, annual independent monitoring of the food supply with regular public progress reporting is recommended to increase accountability and compliance [20], and to ensure progress within an adequate timeframe. In addition, whilst reformulation is important, this should be just one component of a multi-faceted national food and nutrition strategy. Australians are consuming excessive quantities of discretionary foods and inadequate amounts of core foods [33]. Reformulation alone will not address the need for a change in dietary patterns. To maximize public health impact by changing consumer behavior and shifting eating patterns towards a healthier diet, reformulation programs should be part of a broader food and nutrition strategy supported by a Commonwealth-funded campaign that promotes healthy eating, in line with current recommendations [38,39].

The strengths of this study include the use of the FoodSwitch database that has extensive product coverage and rigorous protocols for data collection and quality assurance [28]. Our study builds on the work of the HFP through the use of three years of comprehensive nutrition data from the FoodSwitch database to generate more feasible and appropriate reformulation targets. These data are from foods and beverages available in Australian supermarkets and may not be representative of what is purchased or consumed. Future studies should investigate the impact of these nutrient reformulation targets and a variety of similar scenarios on population nutrient intakes and health outcomes using sales-weighted data. A potential limitation is that the analyses rely on manufacturer reported data on the nutrition information panel, which are the average value for each food product but may have a degree of error. In addition, there may still be a few duplicates included in the dataset due to some products potentially having different barcodes in different years and not being identified through our three-step matching process. However, if the product barcode was changed, it is likely that the product was modified. In addition, this study did not consider technical and food safety aspects of reformulation independently but used the HFP criteria for determining if a target was feasible (if approximately one-third of products already meeting the proposed target, it is likely feasible for the remaining two-thirds). Further research investigating the feasibility of these targets should be conducted.

## 5. Conclusions

This analysis of FoodSwitch data revealed that the majority of proposed HFP targets are too conservative. The opportunity exists to improve the food supply and strengthen the HFP’s population health impact by adopting these more ambitious targets, with further step-wise reductions. This reformulation strategy should be implemented in conjunction with a multi-faceted national food and nutrition strategy and clear implementation, monitoring and evaluation procedures.

## Figures and Tables

**Figure 1 nutrients-12-01346-f001:**
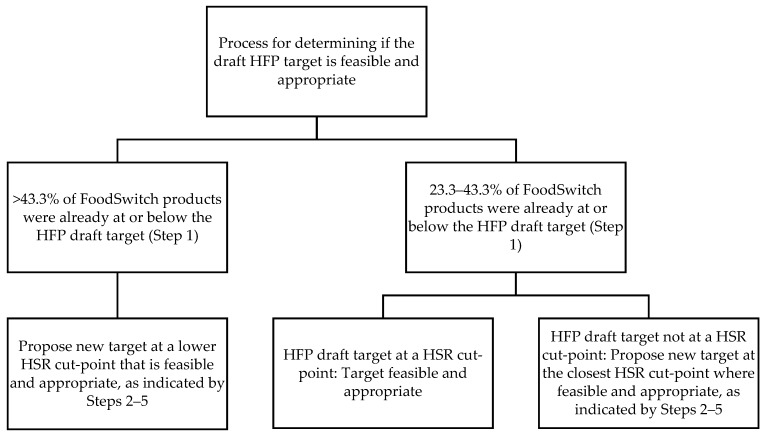
Process for assessing if draft Healthy Food Partnership (HFP) targets are feasible and appropriate. HSR: Health Star Rating.

**Table 1 nutrients-12-01346-t001:** Sodium levels of products included the FoodSwitch and FoodTrack databases, potential reformulation targets, and recommended targets based on FoodSwitch data.

	FoodSwitch			Potential Targets			Percent FoodSwitch products Already Meeting Draft HFP Target (%)	Recommended Target (FoodSwitch; mg/100 g)
HFP Category	Number of Products	Mean (Range; mg/100 g)	Food Track Mean (Range; mg/100 g)	FoodSwitch 33rd percentile (mg/100 g)	UK Target (mg/100 g)	Draft HFP Target (mg/100 g)		
Bread	744	402 (1−1208)	501 (190−934)	377	450	380	35.2%	360
Flat bread	197	531 (22−1200)	649 (342−1000)	400	450	450	44.4%	360
Ready to eat breakfast cereals	710	142 (1−820)	284 (3−785)	28	400	360	88.9%	270
Cheddar and cheddar style variety cheese products	417	681 (1−1960)	711 (550−962)	635	800	710	77.9%	630
Processed cheeses	74	1159 (400−1950)	1208 (700−1600)	880	800	1270	52.7%	810
Crumbed and Battered Proteins: Meat and poultry	228	485 (77−1080)	247 (130−920)	403	380	450	44.7%	400
Crumbed and Battered Proteins: Seafood	217	374 (115−921)	302 (110−815)	290	380	270	29.0%	Same
Gravies and Sauces: Gravies and finishing sauces	287	391 (14−1100)	472 (330−605)	315	450	450	66.6%	360
Gravies and Sauces: Pesto	59	743 (173−1350)	1251 (1251)	569	650	720	54.2%	540
Gravies and Sauces: Asian-style sauces	138	1404 (130−8460)	1371 (258−3147)	667	370	680	34.1%	630
Gravies and Sauces: Other savoury sauces	636	505 (0−8480)	516 (389−683)	316	370	360	48.0%	Same
Pizza	159	545 (115−1230)	504 (350−607)	470	500	450	30.8%	Same
Processed Meat: Ham	111	1075 (1.7−4320)	1308 (1250−1400)	986	Average 650	1005	40.5%	900
Processed Meat: Bacon	124	1110 (430−2900)	1658 (1189−2300)	997	Average 1150	1005	41.9%	Same
Processed Meat: Processed deli meats	108	807.4 (3−1500)	919 (760−1200)	732	650	720	27.8%	Same
Processed Meat: Frankfurts and Saveloys	30	1066 (546−1400)	1050 (1050)	995	750	900	26.7%	Same
Ready Meals	1025	303 (0−1070)	293 (182−605)	244	380	250	36.6%	Same
Sausages	186	682 (79−1840)	690 (355−1050)	581	550	540	23.7%	Same
Savoury Biscuits: Plain savoury crackers and soda biscuits	218	563 (0−1310)	727 (364−1310)	459	700	630	59.2%	450
Savoury Biscuits: Plain corn, rice and other cakes	34	178 (0−608)	392 (118−800)	11	N/A	270	85.3%	<90
Savoury Biscuits: Flavoured biscuits, crackers and corn cakes	454	684 (0−2500)	768 (660−1141)	540	700	720	59.7%	540
Savoury Pastries: Dry pastries	42	544 (270−930)	561 (455−667)	490	450	500	40.5%	450
Savoury Pastries: Wet pastries	195	403 (112−717)	467 (409−603)	362	450	360	32.3%	Same
Savoury Snacks: Potato snacks	296	571 (7−1460)	610 (43−1546)	499	580	500	35.8%	450
Savoury Snacks: Salt and vinegar snacks	44	958 (427−1950)	1010 (670−1390)	760	1000	810	40.9%	Same
Savoury Snacks: Extruded snacks	93	847 (5−1776)	708 (436−1240)	682	800	720	37.6%	Same
Savoury Snacks: Corn snacks	247	440 (5−1820)	366 (1−680)	330	800	360	41.3%	Same
Savoury Snacks: Vegetable, grain and other snacks	190	630 (1−3100)	777 (503−1449)	480	800	450	30.5%	Same
Soups	543	280 (3−887)	350 (149−690)	260	250	270	42.5%	Same
Sweet Bakery: Cakes, muffins and slices	628	294 (8−930)	317 (92−511)	224	280	360	68.2%	270

**Table 2 nutrients-12-01346-t002:** Saturated fat levels of products included the FoodSwitch and FoodTrack databases, potential reformulation targets, and recommended targets based on FoodSwitch data.

	FoodSwitch			Potential Targets			Percent FoodSwitch Products Already Meeting Draft HFP Target (%)	Recommended Target (FoodSwitch; g/100 g)
HFP Category	Number of Products	Mean (Range; g/100 g)	Food Track Mean (Range; g/100 g)	FoodSwitch 33rd Percentile (g/100 g)	UK Target	Draft HFP Target		
Pizza	159	3.9 (1.5−14.9)	4.4 (2.6−5.7)	3.3	N/A	4 g/100 g	68.6%	3
Processed Meat: Frankfurts and Saveloys	30	6.2 (3.4−10.4)	4.8 (4.8)	4.5	N/A	10% reduction across products with saturated fat levels exceeding 6.5 g/100 g	53.3%	4
Sausages	186	6.8 (0.7−17)	7.0 (0.3−11.3)	4.8	N/A	7 g/100 g	50.5%	5
Savoury Pastries: Dry pastries	42	7.2 (2.5−9.4)	6.9 (3.5−8.3)	7.1	N/A	7 g/100 g	31.0%	Same
Savoury Pastries: Wet pastries	195	6.1 (1.1−14)	6.9 (4.3−10.6)	5.4	N/A	7 g/100 g	71.8%	5

**Table 3 nutrients-12-01346-t003:** Sugar levels of products included the FoodSwitch and FoodTrack databases, potential reformulation targets, and recommended targets based on FoodSwitch data.

	FoodSwitch			Potential Targets			Percent FoodSwitch Products Already Meeting Draft HFP Target (%)	Recommended Target (FoodSwitch)
HFP Category	Number of Products	Mean (Range)	Food Track Mean (Range)	FoodSwitch 33rd Percentile	UK Target	Draft HFP Target		
Ready to eat breakfast cereals	710	17.5 (0−44.3) g/100 g	19.2 (1.1−46.0) g/100 g	14.4 g/100 g	20% reduction (12.3 g)	A 10% reduction in sugar across defined products containing over 25 g sugar/100 g, and a reduction in sugar to 22.5 g/100 g for products between 22.5 and 25 g sugar/100 g.	76.8% meeting 22.5 g/100 g	13.5 g/100 g
Flavoured Milk: Mammalian milks	232	9.1 (0.1−22.9) g/100 mL	9.4 (8.3−10.6) g/100 mL	9 g/100 mL	No target if >75% milk	9 g/100 mL	42.7%	Same
Flavoured Milk: Dairy alternatives	61	4.6 (0.1−12) g/100 mL	5.8 (5.8−5.8) g/100 mL	2.9 g/100 mL	No target if >75% milk	4 g/100 mL	41.0%	3 g/100 mL
Muesli bars	273	22.9 (1.8−50.7) g/100 mL	25.3 (14.7−39.8) g/100 mL	18.9 g/100 mL	20% reduction (26.2 g)	A 10% reduction in sugar across defined products containing over 28 g sugar/100 g, and a reduction in sugar to 25 g/100 g for products between 25 and 28 g sugar/100 g.	64.8% meeting 25 g/100 g	18 g/100 mL
Beverages: Soft drinks	505	9.8 (0−34.4) g/100 mL	10.2 (6.2−14.7) g/100 mL	9.5 g/100 mL	Tax applied if >5 g/100 mL	10% reduction in total sugar for products above 10 g sugar/100 mL	42.0% meeting 10 g/100 g	9 g/100 mL
Beverages: Flavoured water, flavoured mineral water, soda water and iced tea	391	4.6 (0−13) g/100 mL	7.7 (4.4−9.3) g/100 mL	3.6 g/100 mL	Tax applied if >5 g/100 mL	5 g/100 mL	61.4%	3 g/100 mL
Sweetened yoghurt	703	11.9 (0−34.8) g/100 g	11.5 (4.0−19.4) g/100 g	10 g/100 g	20% reduction (11.0 g)	13.5 g/100 g	66.9%	9 g/100 g

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
