# Peer review of "Assessing the Healthy Food Partnership’s Proposed Nutrient Reformulation Targets for Foods and Beverages in Australia"

_nutrients, 2020, doi:10.3390/nu12051346_

Round 1

Reviewer 1 Report

Rosewarne and coll. reviewed the nutrient reformulation targets for sodium, sugar and saturated fat proposed by the  Australian Commonwealth Government through the Healthy Food Partnership (HFP) to assess their feasibility and appropriateness. This assessment was performed using Food Switch, a different much larger nutrient composition database built up for the same time period (2015-8). The authors mapped the products listed in the Food Switch database to the proposed HFP targets, calculated the proportion of products already meeting the respective target and sought to determine whether the proposed target nutrient levels were appropriate or a more stringent target was appropriate and feasible. They reported that only half of the proposed HFP targets for sodium and a minority of the  targets for sugar and saturated fat were satisfactory whereas all the others appeared too conservative in as much as over 40% of the products in the respective food categories already met the HFP target according to Food Switch. In conclusion, according to the authors, it would be the case to strengthen the HFP’s population health impact by adopting more ambitious nutrient reformulation targets if the 30 per cent reduction in the average population salt intake objective has to be met by 2025.

This is a well written article which should certainly meet the interest of the Nutrient readership in as much as it deals with a major issue in the area of nutritional prevention of NCDs. The objective of the study is clear and methods reliable and well explained. The data presentation is fine and substantially exhaustive.

I have a general methodological question: in any given food category there may be a few products which exert the greatest impact because of their higher sale volumes compared to most others. Do the authors agree that Ideally, when defining salt (or sugar) targets, it would be important to weigh the sodium concentration of a given product for its sale volume? I wonder whether the authors considered this point, whether they have access to sale volume data and, if not, I would make the advice to mention this limitation in the Discussion.

A few other suggestions that might hopefully further improve the presentation:

-          In the Introduction, it might be the case to explicitly mention the 30% salt reduction WHO/UN objective to be met by 2025;

-          In all the Tables, it would be informative to add the median values (and interquartile range) of nutrient content for the various food categories in addition to the mean content (perhaps the range of individual values may be deleted);

-          In the Discussion section (lines 207-12), it may be appropriate to discuss to what extent the HFP targets for salt and the more stringent targets proposed in this article by the authors are functional to the achievement of the WHO 30% salt reduction goal by 2025.

Reviewer 2 Report

Overall, this is a useful study for assessing the nutrition content of the food supply relative to targets set under an initiative to improve healthiness of the food supply. The analysis makes use of a novel dataset from FoodSwitch. In general, I’m in agreement with the approach of the analysis, but I think it’s important to caveat that reformulations to meet individual nutrition criteria in isolation may not be feasible for some foods either for sensory or food safety reasons. The challenges are even greater if any food categories have multiple targets such as for sodium and saturated fat. Altering a product to achieve multiple criteria might result in unpalatable products that food manufacturers are unable to sell to consumers.

Lines 52-54: Is it possible to cite statistics on where the population stands on saturated fat consumption as well?

Line 69: Are the 63,000 products in FoodSwitch individual barcoded products (with multiple package sizes for a particular formulation) or unique formulations?

Also, for the 8000 products in the analysis, are you able to calculate the relative market share for the product categories in the HFP?

It’s surprising that such a low proportion of the products are covered by the HFP targets. Are there plans to increase the number of categories?

In the introduction, I suggest referencing an article that also simulated the effects of reformulation published in Nutrients last year: Muth, M.K., Karns, S.A., Mancino, L., & Todd, J.E. (2019). How much can product reformulation improve diet quality in households with children and adolescents? Nutrients, 11(3), 618; https://doi.org/10.3390/nu11030618.

Lines 76-77: It would be helpful to have a couple of introductory sentences to guide the reader through the materials and methods section.

Lines 87-88: Do the regulations for nutrition labeling in Australia allow a certain degree of error in the numbers printed on the label? It might be helpful to state just to raise awareness that all of the values are really just estimates.

Lines 95-101: Perhaps this is where you are clarifying that products packaged in multiple product sizes are included in the analysis dataset only once? If so, it needs to be clarified a bit more in the text.

Line 101: How many products were excluded because a nutrient value was missing?

Line 103: How many products were excluded because they couldn’t be mapped to a proposed HFP target?

Line 112: I suggest starting this section with the introduction to Figure 1 first. It’s much easier to follow.

Line 135: The criteria for revising a target based on 23.3% to 43.3% of products already meeting a target seems arbitrary. Is there any basis for establishing this as the criteria? If not, I think it’s important to note this as a limitation of the study. It may also be useful in the future to conduct simulations assuming different criteria.

Lines 143-145: I suggest putting the word “food” in front of “categories” to make it clearer. Also, it would be helpful to have a bit more introductory text to guide the reader through the results section.

Lines 213: Do you have any sense of how these data compare to those that are scanner data available from commercial data suppliers such as Nielsen or Kantar? In the US, the overall count of food barcodes seems to be around 700,000. Are there other sources for Australia that provide an overall estimate of the number of food barcodes to provide a sense of the count coverage of FoodSwitch data. Also, scanner data for the foods in the analysis would allow you to assess the relative sales volumes of foods that do and don’t meet the criteria.
